# Investigation of the Effect of Atmospheric Plasma Treatment in Nanofiber and Nanocomposite Membranes for Piezoelectric Applications

**DOI:** 10.3390/membranes13020231

**Published:** 2023-02-14

**Authors:** Papia Sultana, Mujibur Khan, Debdyuti Mandal, Mohammadsadegh Saadatzi, Sourav Banerjee

**Affiliations:** 1Mechanical Engineering Department, Georgia Southern University, Statesboro, GA 30458, USA; 2Integrated Material Assessment and Predictive Simulation Laboratory, Department of Mechanical Engineering, University of South Carolina, Columbia, SC 29208, USA

**Keywords:** nanofiber, nanocomposite, membrane, atmospheric plasma, piezoelectricity

## Abstract

In this work, we report the effect of steady-state atmospheric plasma (Corona discharge) in nanofibers and nanocomposite membranes for piezoelectric applications. The investigation was performed in PVDF (Poly vinylidene fluoride) nanofibers, CNT (Carbon Nanotubes)-reinforced PVDF nanocomposites, and PAN (Poly acrylonitrile) nanofiber membranes. Steady-state plasma was generated with a high voltage power source with 1 mA discharge current output and 6 kV discharge voltage, and the gap between tip and the material was maintained to be 1 cm. For the fabrication of nanofibers and nanocomposite membranes, an electrospinning method was used. The electrospinning parameters, such as flow rate and voltage, were optimally tuned for obtaining uniform nanofibers and nanomembranes. Along with the plasma treatment, heat treatment above the glass transition temperature was also conducted on the nanofiber membranes. Using a Scanning Electron Microscope (SEM), the morphology of the nanofibers was observed. X-ray Diffraction (XRD) demonstrated the polycrystallinity of the nanofibers. Fourier Transform Infrared Spectroscopy (FTIR) analysis of the PVDF nanofibers shows a peak at 796 cm^−1^ representing α-phase (C-H rocking) in the control sample which is absent in the treated samples. Raman spectroscopy of PVDF nanofibers identifies a Raman shift from 873 cm^−1^ to 877 cm^−1^ (denoting β-phase) for plasma-treated samples only. Electron Paramagnetic Resonance (EPR) concludes that the intensity of the free radicals increases from 1.37 to 1.46 (a.u.) after plasma treatment. Then, sensors were fabricated from the PVDF nanofibers, MWCNT-reinforced PVDF nanofibers, and PAN nanofibers to characterize their piezoelectric properties. The impact test results showed that the atmospheric plasma and heat-treated samples had 86%, 277%, and 92% increases of the d_33_ value (piezoelectric coefficient) in the case of PVDF nanofibers, MWCNT-reinforced nanofibers, and PAN nanofibers, respectively. It was also observed that the capacitance of the nanofiber membranes has increased due to the plasma treatment.

## 1. Introduction

Poly(vinylidene fluoride) [PVDF;(CH_2_CF_2_)n] is a semi-crystalline and ferroelectric polymer. Due to its chemical resistance, thermal stability, high mechanical strength, large remnant polarization, short switching time, and unique electrical properties, PVDF is considered for applications in organic electronics, biomedicine, optoelectronics, energy harvesters, etc.

The piezoelectric effect is observed in certain types of crystalline materials, which is defined as the linear electromechanical interaction between the mechanical and electrical states of the materials. Due to the biocompatibility, biodegradability, low cost, and low power consumption characteristics, piezoelectric polymers are a suitable choice over any other piezo materials. Piezoelectric properties of polymers such as Polyvinylidene fluoride (PVDF), Polyacrylonitrile (PAN), Polyimide, Polyvinylidene Chloride (PVDC), etc., have various significant applications as energy harvesters, nanogenerators, sensors, actuators, piezoelectric motors, etc. [1,2,3,4,5]. In this study, the piezoelectric property of electroactive polymer nanofibers like PVDF and PAN is explored for energy harvesting purposes [6].

In PVDF, the amorphous and crystalline phases coexist. Generally, PVDF has three major polymorph phases (α-, β- and γ-). The most common crystalline phases observed in PVDF are the α- and β-phases. The α-phase of PVDF is a non-polar, electrically inactive phase, and it is also the most stable phase [7,8], when PVDF is prepared by cooling from the melt. The β-phase exhibits the strongest ferro-, piezo-, and pyroelectric properties, due to its large spontaneous polarization. This phase is generally obtained through uniaxial or biaxial stretching of crystallization from solution under a special condition or through the application of high electric fields to PVDF in its α-phase. Another widely used polymeric material, PAN, has high resistance to chemicals and a high adsorption property for heavy metal ions [9,10,11].

In this work, we have investigated the effect of a direct current-driven corona discharge on electroactive polymer nanofibers and CNT-reinforced nanocomposite membranes to enhance the piezoelectric properties. In comparison with other conducting nanofillers, CNT has higher mechanical strength and elasticity. Moreover, it has been reported that a graphene sheet demonstrates surface piezoelectricity and flexoelectricity in certain cases such as: (I) when non-centrosymmetric pores are formed in it [12,13,14], (II) if a bending moment and biaxial strain [15,16,17] is applied to it, and (III) through the selective surface adsorption of atoms [13,18,19,20]. Also, it is demonstrated that a significant bending moment is initially formed in carbon nanotubes (CNTs), which can lead to the manifestation of the surface piezoelectricity [14]. For these reasons, in this work we wanted to investigate the piezoelectric properties of MWCNT by fabricating the MWCNT-reinforced PVDF nanofibers.

During the corona treatment, a high voltage is applied from an electrode tip to the surface of the sample. When this discharge passes through the air gap, the generated electrons in the discharge impact the surface with a high energy, which breaks the molecular bonds on the surface. Due to the breaking of the molecular bonds, free radicals are generated. These free radicals form various chemical functional groups in the surface. These groups increase the surface energy, which modifies the surface characteristics [21,22]. This phenomenon is observed to exist only in nanoscale materials, whereas macroscale materials do not exhibit any significant change of corona discharge effect. This study is a strong support regarding the effectiveness of the corona treatment to induce piezoelectric properties on the electrospun polymer nanofibers of PVDF and PAN. The nanofibers are fabricated by the electrospinning process. After the corona treatment, the electrospun nanofiber membranes are used to fabricate sensors to quantify the output voltage during an impact with certain amount of force. These corona-treated nanofiber membranes can be used to produce multilayer piezoelectric sensors, and such smart sensors can be further used in a structural health monitoring system [23,24,25,26]. Serving as a baseline for future work, our goal is to scale up the fabrication of novel polymer nanofiber composites with the enhanced piezoelectric property for energy harvesting purposes.

## 2. Experimental Section

### 2.1. Materials

Polyvinylidene Fluoride (PVDF) powder, Polyacrylonitrile (PAN) powder, and *N*,*N*-Dimethylformamide (DMF) solvent were procured (Sigma Aldrich, St. Louis, MO, USA). DMF is used as the solvent system for PVDF and PAN. The average density, melting point, and molecular weight of PVDF are approximately 1.74 g/mL, 171 °C, and 534,000 g/mol respectively. The average density, melting point, and molecular weight of PAN are approximately 1.18 g/mL, 317 °C, and 150,000 g/mol respectively. Multi-walled Carbon Nanotubes (MWCNT) (10–20 nm in diameter and approximately 10–30 µm in length) were obtained from Cheaptubes Inc.,Cambridgeport, VT, USA. The following four categories of samples were prepared: (a) PVDF Film; (b) PVDF Nanofiber membrane; (c) PVDF + 0.2% MWCNTs Nanofiber membranes; and (d) PAN nanofiber membranes. All the samples were treated with heat and atmospheric plasma (corona).

Table 1 shows the types of materials and post-processing techniques used for the experiment.

### 2.2. Polymer Solution and Film Preparation

For the electrospinning of polymer nanofibers, the following three mixtures were prepared: (a) Only PVDF solution, (b) MWCNT with 0.2% *w*/*w* of PVDF solution, and (c) Only PAN solution. For fabricating pristine PVDF nanofiber membranes, PVDF was added to 10% *w*/*v* with DMF solvent, and then it was magnetically stirred for 1 h to make a homogenous solution. For fabricating MWCNT-reinforced PVDF nanofiber membranes, MWCNTs with 0.2% *w*/*w* of PVDF and 15% *w*/*v* PVDF were dissolved in DMF solvent. Firstly, MWCNTs of 0.2% *w*/*w* of PVDF were dispersed ultrasonically in DMF solution at 20% amplitude for 10 min with 1 min of a pulsating interval by a Sonics Vibra Cell Model CV 18 ultrasonic liquid processor (Sonics & Materials, Inc., Newtown, CT, USA). During the process, an ice bag was used to surround the beaker to avoid overheating and evaporation of the solution. Then, PVDF was added into the dispersed MWCNT-DMF solvent mixture and stirred using a magnetic stirrer for 1 h until a uniform mixture was observed. For pristine PAN nanofiber membranes, PAN was added to 5% *w*/*v* with DMF solvent and dissolved by a magnetic stirrer with continuous heating at 50 °C for 2 h.

Additionally, to investigate the inherent piezoelectric property of PVDF, PVDF films were fabricated to compare with PVDF nanofibers. To fabricate films, PVDF was first added to 20% *w*/*v* with DMF solvent and stirred for 1.5 h. Then, the solution was spread over on top of aluminum foil to make a very thin layer. Then, the layer was dried by heating on a hotplate for 40 min at 50 °C, and afterwards peeled from the foil.

### 2.3. Electrospinning Process and Parameters

Polymer nanofibers were fabricated by the electrospinning process performed using a NF-500 Electrospinning Unit (MECC, Fukuoka, Japan). The inner chamber of the NF-500 unit is connected to a dehumidifier unit, which maintains a constant humidity level during electrospinning. Utilizing two feed pumps with controlled feeding, two syringes of solutions filled with polymer are connected to the nozzle of a spinneret. The single nozzle spinneret assembly includes a housing for placing a 6 mL syringe filled with a polymer solution. A 27-gauge needle is attached to the tip of the syringe. The positive terminal of a high voltage DC power supply with capacity up to 60 kV is connected to the spinneret and the ground terminal is connected to the collector plate, which is placed at a specific distance from the spinneret as can be seen in Figure 1. The electric field strength can be modified by adjusting the distance between the spinneret nozzle tip and the collector plate. Aluminum foil, which is used to cover the grounded plate collector, functions as a conductive substrate for the deposition of the nanofibers during electrospinning. By using the control unit of the NF-500 system, spinneret tip-to-collector distance (TCD) and feed rates of polymer solution can be adjusted. Fabricating different types of nanofibers by the electrospinning method requires specific electrospinning parameters, which are obtained by experimentation process. The electrospinning parameters for fabricating PVDF nanofibers are as follows: 1.2 mL/h flow rate; 30 kV voltage; 150 mm TCD; 10 mm/s traverse speed; and 120 mm traverse distance. For fabricating the MWCNT-reinforced PVDF, electrospinning parameters are given as follows: the flow rate 0.9 mL/h, the voltage 28 kV, and TCD 150 mm. Then, the electrospinning parameters used for fabricating pristine PAN nanofibers are as following: flow rate 0.7 mL/h, voltage 31.5 kV, and TCD 145 mm.

### 2.4. Post-Processing Treatments

Three sets of samples were prepared from electrospun pristine PVDF nanofibers, MWCNT reinforced PVDF nanofibers, pristine PAN nanofibers, and PVDF film. These sample sets were classified based on their post-processing surface treatments.

Heat treatment of the nanofibers is performed by using Master ProHeat Professional Heat Gun (Master Appliances Corp, Racine, WI, USA), and simultaneously, the surface temperature is measured by Greenpro GP0145 Infrared Thermometer Temperature Gun. The whole surface of the nanofibers is heated up to 80 °C and held for approximately 1 h to reach above its “Glass Transition Temperature” that leads to the change in conductivity. The glass transition temperature is the temperature where polymer materials change from a rigid glassy material to soft (not melted) material with increased molecular mobility. This increased molecular mobility results in significant changes in the properties of the material.

Stable corona is generated in a high voltage setup with voltage as a constant parameter and current being varied by changing the anode and cathode distance, as shown in Figure 2. The needle tip where plasma is generated acts as the anode, and the needle is connected to a high-power supply. Aluminum foil acts as the cathode, which is connected to the ground. Nanofibers are kept on top of the aluminum foil, between the anode and cathode, to be treated by plasma for 2 h to expose both sides of the membrane surface to the applied plasma. The distance between the anode and cathode is kept stable, so that a 1 mA current is generated with around 6 kV voltage. The air gap between the tip and material surface is kept at around 1 cm.

To determine the corona and heat treatment effect cumulatively, on one sample set, heat treatment is done for 1 h. After that, the sample is cooled down to room temperature, and then the corona treatment is performed on nanofiber’s surface for 2 h.

### 2.5. Impact Testing Process

All the treated and pristine PVDF, MWCNT-reinforced PVDF, pristine PVDF film, and pristine PAN membranes were used to make sensors to record the response during direct impact. First, the nanofiber membranes were cut into rectangular shapes and placed between a silicone layer. Then, adhesive copper tape and copper wirings were used to create the electrical connections which is shown in Figure 3. To measure the applied force, off-the-shelf pressure sensors were connected to some sensors to ensure that exact amount of force was applied during the impact. Following is the image of sensors:

Impact tests were done for testing the piezoelectricity and the piezoelectric characterization of all the polymer specimens. During impact, the stress waves were generated and propagated through the fabricated sensors. All polymer sensors were integrated with an impact sensor to measure the impact force. Applied force on the samples was approximately 0.65 N at 0.3048 m impact height. An oscilloscope was used to receive signals from the sensors. Through copper wires, sensors were connected to the oscilloscope to record the output during impact loading. Using the signal responses of treated sensors, their piezoelectric coefficients were calculated. The sensors were attached to the rectangular metallic box. When a ball with specified weight was dropped on to the sensor, a peak was observed in the oscilloscope. A circular glass plate was used for applying uniform force on the membrane sensors. The impact test setup is shown in Figure 4.

The value of piezoelectric coefficient, g_33_, is 0.393 Vm/N. The output voltage can be determined as:V = −g_33_ (F_33_/Area) × t = g_33_ T_33_ t [V](1)
where, Area = 7.85 × 10^−5^ [m^2^], thickness t = 430 [µm], g_33_ = 0.393 Vm/N.

After getting the output voltage from oscilloscope, we calculate the d_33_ parameter to quantify the piezoelectric property of the membrane by the following equation:d_33_ = α (C_PVDF_ V/F_33_)(2)
where, calibration constant from PZT-5H α = d_33 PZT_/d_33 exp_PZT_ = 593 [pC/N]/293 [pC/N] = 2.024, fabricated sensor’s capacitance C_PVDF_ = 2.3 nF, out-put voltage = V, applied impact force F = 0.65 N.

## 3. Results and Discussions

### 3.1. Scanning Electron Microscope (SEM)

The morphology and fiber diameter were observed on a Field Emission Scanning Electron Microscope (model number JEOL JSM 7600F, Peabody, MA, USA). The morphology of pristine PVDF membrane, CNT reinforced PVDF membrane, and PAN nanofiber membrane is observed. During imaging, the acceleration voltage was varied from 5 kV to 20 kV using a secondary electron detector. Before imaging, the samples were coated with a thin homogenous Au layer by an ion sputterer named DESK V Sputter machine (Denton Vacuum, Moorestown, NJ, USA). Sputtering is essential because the samples are not conductive, so the resulting secondary electron signal is feeble for topographic examinations in the SEM.

The SEM images of the pristine PVDF membranes are in Figure 5. The pristine PVDF is translucent in nature. The top surface has mixed topography with pores and a rough layer surrounding it. Fibers are of varying sizes from 100 nm to 300 nm with occasional beads.

The SEM images of the MWCNT-reinforced PVDF membrane can be seen in Figure 6. The acicular nanotubes were randomly dispersed and no nanotubes were seen outside of the fibers. Unlike the PVDF nanofibers, PVDF-MWCNT nanofibers are more of a circular pattern. The fiber size varies from 100 nm to 200 nm, however, the bead size is observed to be as large as 500 nm. Surface crazes are observed on the fiber surface, but no voids have been observed. Figure 7 shows SEM images of the PAN nanofibers. The fibers’ diameter is more uniform, around 90 nm, with less or no surface craze and pores. Surface textures are smooth compared to the PVDF nanofibers. The cross sections are circular and uniform. The SEM pictures in all these categories reveal that random nanofibers orientation created nanosized contact junctions. No such fiber structure and nanojunctions are present in the PVDF film since the film was solidified from molten pool. In a previous study [27], it was reported that nano-contact junctions of CNTs played a significant role in the contribution to the thermoelectric effect by increasing the charge carrier concentration. In the impact test results provided in the later sections of this paper, we observed that the piezoelectric coefficient (d_33_) has increased significantly from PVDF film to PVDF nanofiber membrane samples when treated with heat and corona discharge. It was maximum with MWCNT-reinforced PVDF samples, possibly due to the presence of nano-contact junctions. The capacitance also increased from film to nanofiber membrane.

### 3.2. X-ray Diffraction (XRD) Analysis

An X-ray diffraction technique has been utilized to detect changes in crystalline and amorphous regions, along with the degree of crystallinity. It is used to measure crystal structure, grain size, texture, and residual stress of materials through the interaction of X-ray beams with samples. As X-rays are predominantly diffracted by electron density, analysis of the diffraction angles can be used to produce an electron density map of a given crystal or crystalline structure.

The polycrystallinity of the pristine PVDF films are investigated using XRD analysis. Figure 8 shows the XRD diffraction pattern of control, corona-treated, heat-treated, and both corona- and heat-treated samples. We observed the peak at 2θ = 16.96° corresponds to cubic α-phase of crystal structure reflection (100) [28].

To observe the α-phase peak carefully, the peak is zoomed in the range of 2θ = 16° to 18°. Figure 9 is focused on the peak only. A right-hand shift is observed in the phase from the control sample to the treated sample.

Then, normalization of the curve is performed to compare each treatment. For that, the maximum intensity of each treatment, which is different from other samples, is taken as 1, so that all the intensity values get a ratio of 1. Then, these values are plotted as in Figure 10.

It can be seen that after heat treatment, the α-phase shifted slightly towards a greater angle, from 16.3° to 17°. This shift is also observed in the sample, which was treated with both heat and corona discharge; however, it is not observed in only corona-treated samples. The heat treatment causes the highest sharp peak with a shift in angle. Thus, we can conclude that the heat treatment effect has the significant modification from α-phase to β-phase in PVDF nanofibers.

### 3.3. Fourier-Transform Infrared (FTIR) Spectroscopy

The fraction of the β-phase presence in PVDF nanofibers is assessed using FTIR spectroscopy with a spectrophotometer Thermo iS10 FT-IR (×2) (ThermoFisher Scientific, Waltham, MA, USA). The characteristic peaks attributed to the FTIR absorbance band of α-phase, electroactive polar β-phase, and semi-polar γ-phase can be identified. During electrospinning, when the polymer solution is placed in a strong electric field, the CF_2_ and CH_2_ dipoles of the PVDF are oriented due to uniaxially stretching by strong electric forces and lead to the formation of all trans conformation. The presence of different phases can be identified by FTIR, as they show a specific characteristic peak.

FTIR of PVDF nanofiber membranes (Figure 11) shows a peak at 796.0 cm^−1^ in the control sample, which belongs to the α phase. This peak is not that prominent in all the treated samples. There is another particular peak at 1233 cm^−1^, which is found in all the samples, but in the control sample, the peak has lesser absorbance value. This is a characteristic peak of gamma phase. Apart from these peaks, there are peaks at 763, 975 cm^−1^ (characteristic peak of α phase) and 840, 1275 cm^−1^ (characteristic peak of β phase) [29,30,31,32].

FTIR of MWCNT-reinforced PVDF membrane is shown in Figure 12. There are peaks at 763, 797, 975 cm^−1^, which are characteristic peaks of α-phase, and also peaks at 840, 1275 cm^−1^, which are characteristic peaks of β-phase. There is an increase of α-phase peaks in CNT-reinforced PVDF samples. This is reflected in the d_33_ property of the CNT-reinforced PVDF samples, as observed from the impact test results. There is another peak at 674 cm^−1^, which has very prominent absorbance values in corona, and corona + heat treated samples, which indicate corona treatment generates this peak.

FTIR results of PVDF sample and MWCNT-reinforced PVDF sample is summarized in Table 2 and Table 3, respectively.

### 3.4. Raman Spectroscopy

Raman spectroscopy was performed in DXR Raman Microscopy. Figure 13 shows the Raman spectroscopy of PVDF nanofiber membrane samples and MWCNT-reinforced PVDF samples. By comparing these two spectra in Figure 13, it can be observed that the characteristic peak of CNT at 1351 cm^−1^ and 1597 cm^−1^ existed only in the MWCNT-reinforced samples.

Figure 14 is the Raman spectroscopy of the control, corona-treated, heat-treated, and corona + heat-treated PVDF nanofiber membranes. It is observed that characteristic α-phase at 788 cm^−1^ existed in all the samples, and there is a shift of this peak from the control to the treated sample. However, this α-phase peak is not prominent in (corona + heat)-treated samples. Two other peaks at 2972 cm^−1^, which corresponds to C–H Alkayl bonding, disappears in the corona + heat-treated samples, but exists in all other samples. Another peak at 2432 cm^−1^ shows a sharp peak only in corona-treated samples.

Figure 15 is the Raman spectroscopy of control, corona-treated, heat-treated, and corona + heat-treated MWCNT-reinforced PVDF samples. There are some characteristic peaks found in Raman at 1598 cm^−1^ (corresponds to G band), 1435 cm^−1^ (corresponds to D band), 1350 cm^−1^ (corresponds to G band), 2688 cm^−1^ (corresponds to G band), and 2979 cm^−1^ (corresponds to G band) [32,33]. These peaks are observed in all the samples. The G-band is an intrinsic feature of carbon nanotubes, that is, related to vibrations in all SP carbon materials, and it is also assigned to the in-plane vibration of the C–C bond. The G band is a tangential shear mode of carbon atoms that corresponds to the stretching mode in the graphite plane. The D-band is activated by the presence of disorder in carbon systems. The G″ band attributed to the overtone of the D band. The shifts observed in the MWCNT-reinforced PVDF are all right-shift, such as right-shift of D band from 1350 cm^−1^ to 1354 cm^−1^ and right-shift of G band from 2688 cm^−1^ to 2690 cm^−1^. From the Raman data analysis, the characteristic β-phase peaks at 614 cm^−1^ (contributes to CF_2_ vibration) and 842 cm^−1^ (contributes to out-of-phase combination of CH_2_ rocking and CF_2_ stretching mode) wavenumbers have also been found.

### 3.5. Electron Paramagnetic Resonance (EPR) Spectroscopy

EPR is a characterization technique to obtain information regarding the chemical structure of radicals. It can also provide information regarding spin-concentration. The EPR experiments were performed at room temperature on an EPR spectrometer using a 300 kHz field attenuation, 30 gauss of amplitude modulation, and an applied microwave power of 2 mW. The spectra were recorded by sweeping the static magnetic field (from 3340 to 3380 G). This test was performed on two categories of samples; one is the control sample, and another one is the corona-treated sample. EPR on the corona-treated sample is performed immediately after the corona treatment. For the experiment, the samples were collected in a glass tube and placed in the spectrometer cavity. It is found from literature [34] that each covalent bond of the –VDF repeating unit, for instance, C–F, C–H and C–C bond, can be broken when the polymer is exposed to corona treatment. There may be alkyl and per-oxy radicals resulting from the C–H and C–F bond breaking [35,36].

The most probable five free radicals [37] expected to form due to corona treatments are (1) –(CF=CH)n–CF–, (2) –CF_2_–CH–CF_2_–, (3) –CH_2_–CF–CH_2_–, (4) –CF_2_–CH_2_, and (5) –CH_2_–CF_2_. Figure 16 and Figure 17 are the results of EPR on the control sample and corona-treated sample.

It is observed from the results that the intensity of free radicals of the PVDF nanofiber membrane is increased from 1.37 (a.u.) to 1.46 (a.u.) after the corona treatment. Therefore, the quantity of free radicals increased by 8% due to corona discharge treatment.

### 3.6. Impact Test Results

The oscilloscope connected with the sensors shows a voltage spike during the impact, which results from the electrical response of polymer nanofibers. A representative voltage response of atmospheric plasma (corona discharge) treated and corona- and heat-treated PAN membrane sensors is shown in Figure 18. However, the control samples, without any treatment, have no output response during the impact. The results of the output voltage, capacitance, and d_33_ value of PVDF films, PVDF nanofiber membranes, MWCNT-reinforced PVDF nanofiber membranes, and PAN nanofiber membranes are shown in Table 4. The voltage response demonstrates that the corona discharge treatment effect on the nanofibers results in higher quantity of electroactive phases.

The comparison of the capacitance of all the film and nanofiber membranes can be seen in Figure 19. It is observed that PVDF nanofiber membranes show significantly higher capacitance compared to PVDF film. Additionally, it shows that capacitance increases with the treatment of corona discharge and heat. Similar phenomena have been observed for PAN nanofiber membranes. However, MWCNT-reinforced PVDF nanofiber membranes show less capacitance when compared with PVDF nanofiber membranes.

The comparison of d_33_ of film and nanofiber membranes can be seen from Figure 20. The d_33_ is observed to be higher in categories of nanofiber membranes when compared with the film. It is also observed that the d_33_ is almost two times greater in corona discharge and heat-treated MWCNT-reinforced PVDF samples than the treated pristine PVDF samples. In all categories of samples, d_33_ shows an increasing trend in heat and corona discharge-treated samples. The maximum output voltage (67 mV) and the d_33_ (102 pC/N) values are found on the MWCNT-reinforced PVDF nanofiber samples when treated with heat and corona discharge. However, maximum capacitance (0.93 nF) was observed in the heat and corona discharge-treated PVDF nanofiber membranes.

One notable finding from the impact test is that the maximum output voltage, capacitance, and the d_33_ parameter are found on the heat + corona-treated samples.

## 4. Conclusions

In this work, PVDF nanofiber membranes, MWCNT-reinforced PVDF nanofibers, and PAN nanofibers were fabricated by electrospinning process and treated with atmospheric steady-state plasma (corona discharge). Sensors were fabricated from these nanofiber membranes and tested. In all categories of samples, the piezoelectric coefficient (d_33_) showed an increasing trend in heat and corona-treated samples. The d_33_ was almost two times greater in corona and heat-treated MWCNT-reinforced PVDF samples than the treated PVDF samples. Treated PVDF film samples exhibited 20 mV output voltage, while the treated PVDF nanofibers showed 65 mV output voltage. The d_33_ was observed to be higher in nanofiber membranes when compared with the PVDF film. XRD results showed that the α-phase found in control sample shifted slightly in all the treated samples towards a greater angle, from 16.3° to 17°. In the FT-IR characterization of PVDF samples, we found a peak of α-phase in 796.0 cm^−1^ in the control sample, which was not prominent in the treated samples. A peak in 1233 cm^−1^ corresponding to the γ-phase was found in all the samples. We also found peaks at 763 cm^−1^ and 975 cm^−1^ corresponding to α-phase and peaks at 840 cm^−1^ and 1275 cm^−1^ corresponding to β-phase in all the samples. There was an increase of α-phase peaks in MWCNT-reinforced PVDF samples compared to the only PVDF samples. In the MWCNT-reinforced PVDF samples, we found peaks at 763 cm^−1^, 797 cm^−1^, and 975 cm^−1^ corresponding to α-phase and 840 cm^−1^ and 1275 cm^−1^ corresponding to β-phase. From Raman spectroscopy, it was observed in the PVDF samples that characteristic α-phase at 788 cm^−1^ existed, and there was a right-shift of this peak from the control to the treated sample. The characteristic peaks of CNT (1351 cm^−1^, 1597 cm^−1^) were observed in MWCNT-reinforced PVDF samples. We also found peaks at 1598 cm^−1^, 1350 cm^−1^, 2688 cm^−1^, and 2979 cm^−1^ corresponding to G-band, a peak at 1435 cm^−1^ corresponding to D-band, and peaks at 614 cm^−1^ and 842 cm^−1^ corresponding to β-phase. All the shifts observed in MWCNT-reinforced PVDF samples were right-shift. In the EPR characterization, we compared the control sample and corona-treated sample and observed an 8% increase of free radicals in the corona-treated sample. Thus, it can be concluded that corona discharge treatment increases the quantity of free radicals, which eventually was evident in the piezoelectric properties of the samples. Hence, all the characterization techniques demonstrated the effect of atmospheric plasma treatment in the enhancement of piezoelectric properties of nanofibers and nanocomposites. Utilizing the piezoelectric properties of PVDF nanocomposites as sensors, a wide range of applications in medical diagnostics, wearable systems, structural health monitoring systems, electromechanical equipment, etc., can be achieved.

## Figures and Tables

**Figure 1 membranes-13-00231-f001:**
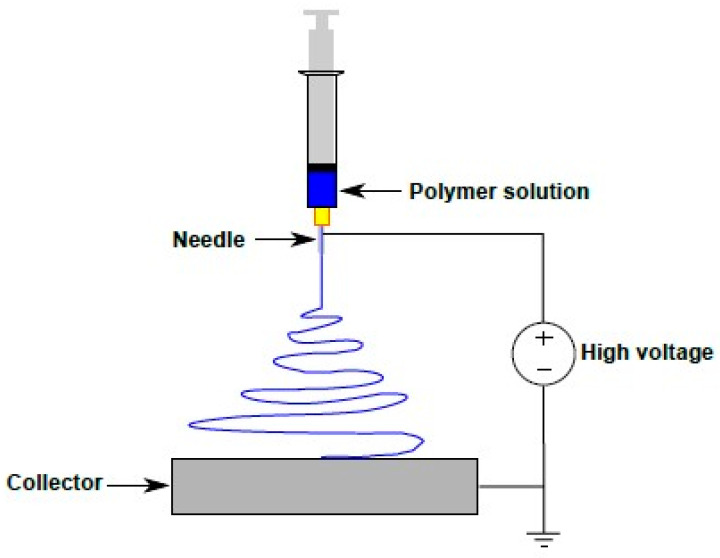
Schematic illustration of far field electrospinning process.

**Figure 2 membranes-13-00231-f002:**
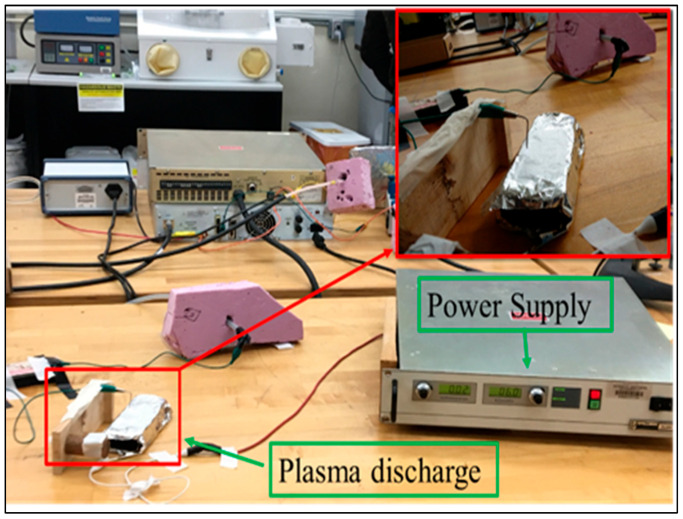
Set-up for corona treatment.

**Figure 3 membranes-13-00231-f003:**
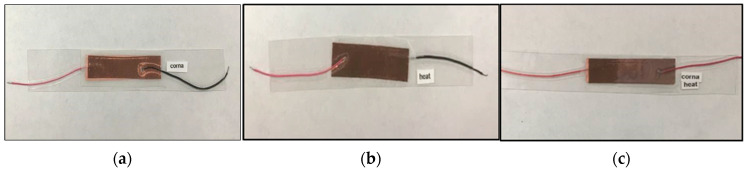
Pristine PVDF membrane sensors: (**a**) Corona treated, (**b**) Heat treated, (**c**) Heat + Corona treated.

**Figure 4 membranes-13-00231-f004:**
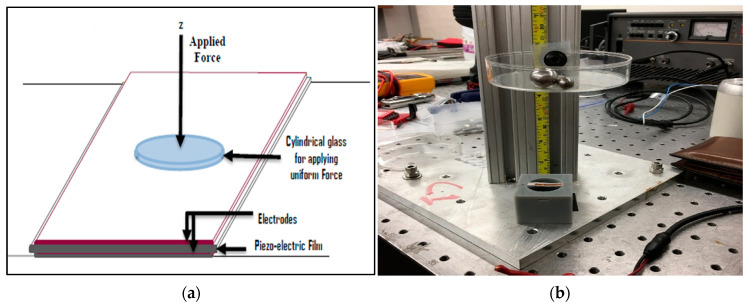
(**a**) Schematic of impact test, (**b**) Impact test setup.

**Figure 5 membranes-13-00231-f005:**
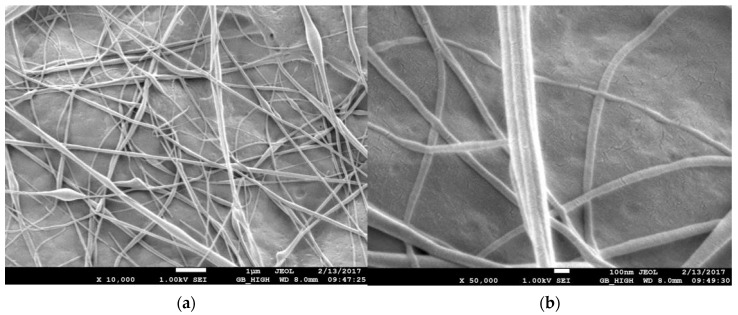
SEM images of pristine PVDF nanofibers (**a**) at 10,000 and (**b**) at 50,000 magnifications.

**Figure 6 membranes-13-00231-f006:**
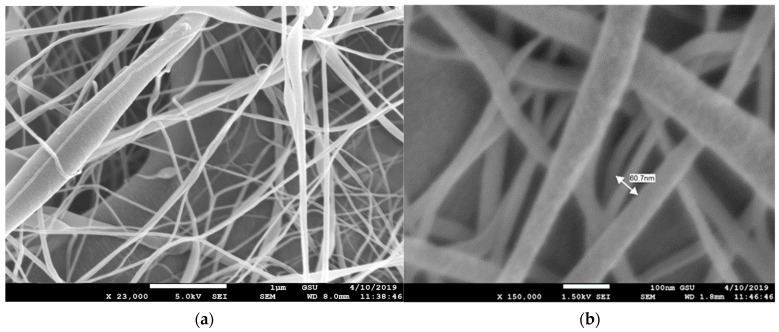
SEM image of MWCNT-reinforced PVDF nanofibers (**a**) at 23,000 magnification and (**b**) at 150,000 magnification.

**Figure 7 membranes-13-00231-f007:**
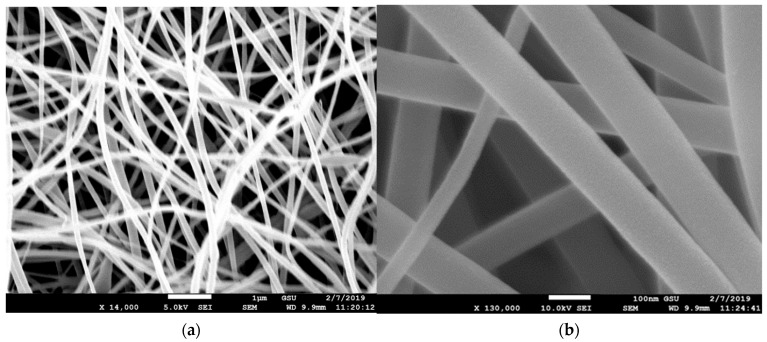
SEM image of pristine PAN nanofibers (**a**) at 14,000 magnification and (**b**) at 130,000 magnification.

**Figure 8 membranes-13-00231-f008:**
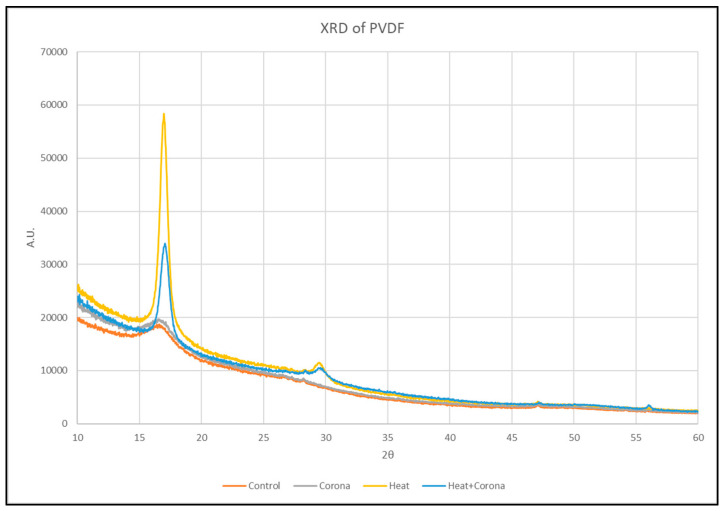
XRD of control and treated PVDF nanofibers.

**Figure 9 membranes-13-00231-f009:**
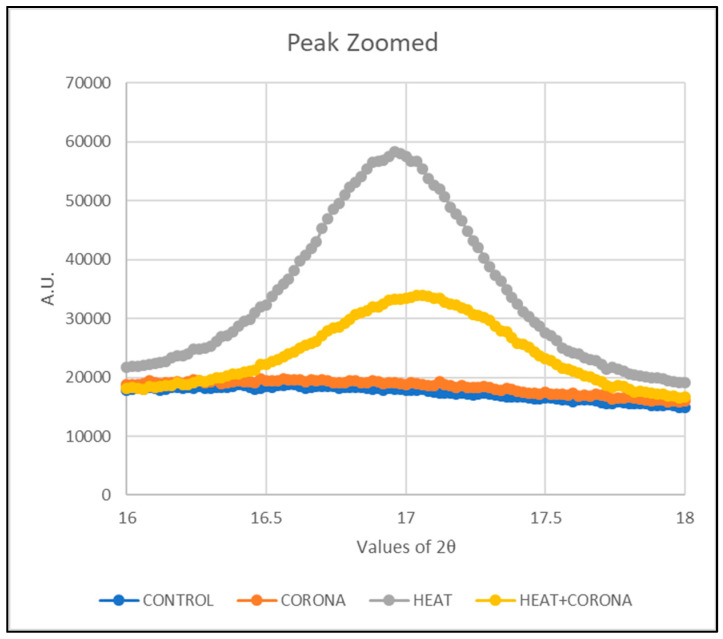
Zoomed in α-phase peak at 16.96° of PVDF nanofibers.

**Figure 10 membranes-13-00231-f010:**
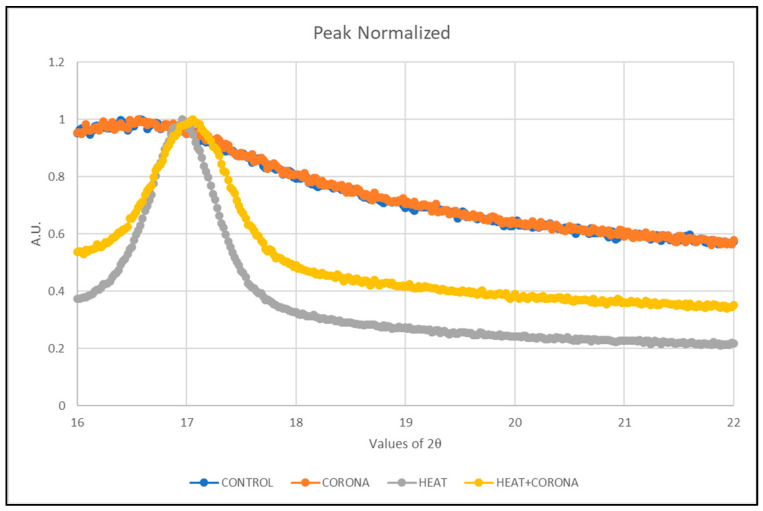
Normalized α-phase peak of PVDF nanofibers.

**Figure 11 membranes-13-00231-f011:**
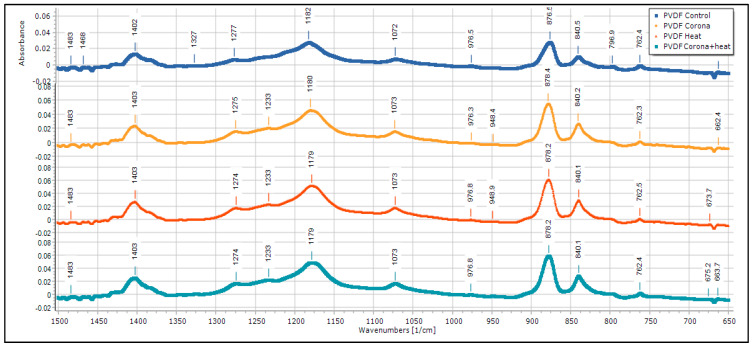
FTIR of pristine PDVF membrane.

**Figure 12 membranes-13-00231-f012:**
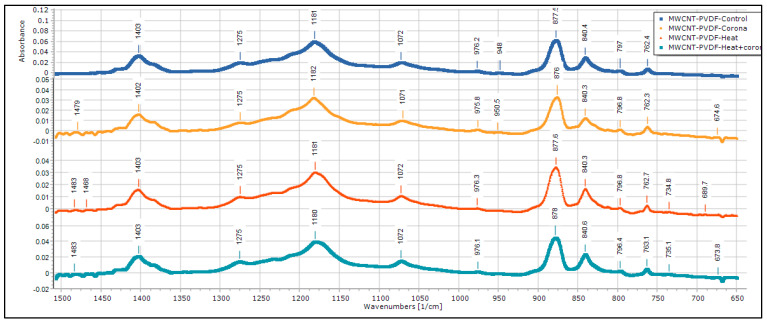
FTIR of MWCNT reinforced PVDF membrane.

**Figure 13 membranes-13-00231-f013:**
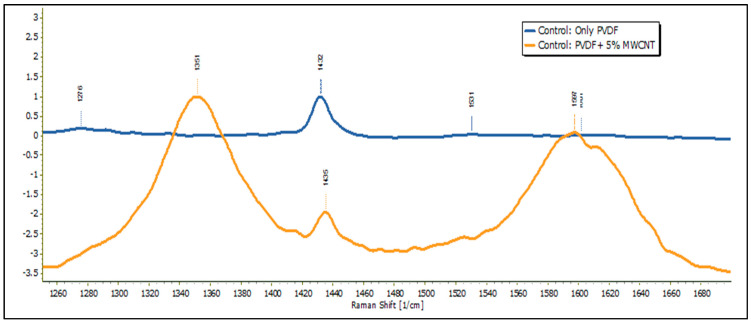
Raman results for Only PVDF and PVDF + 5% MWCNT sample.

**Figure 14 membranes-13-00231-f014:**
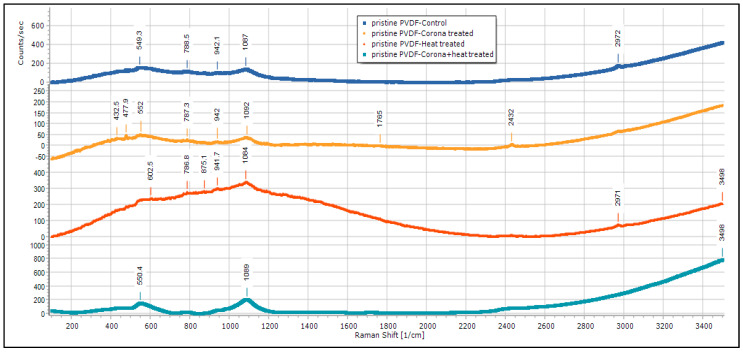
Raman results for pristine PVDF membrane.

**Figure 15 membranes-13-00231-f015:**
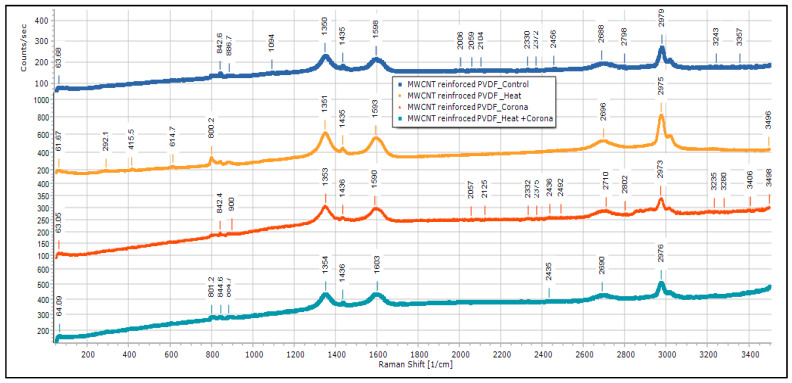
Raman results for MWCNT-reinforced PVDF membranes.

**Figure 16 membranes-13-00231-f016:**
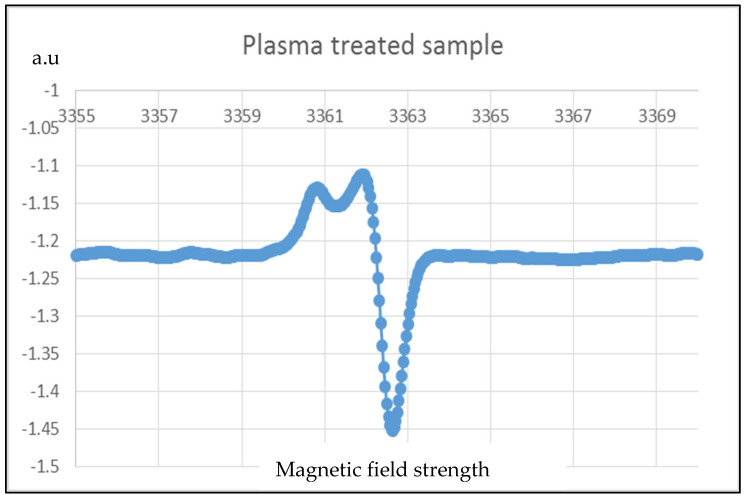
EPR results for corona-treated PVDF sample.

**Figure 17 membranes-13-00231-f017:**
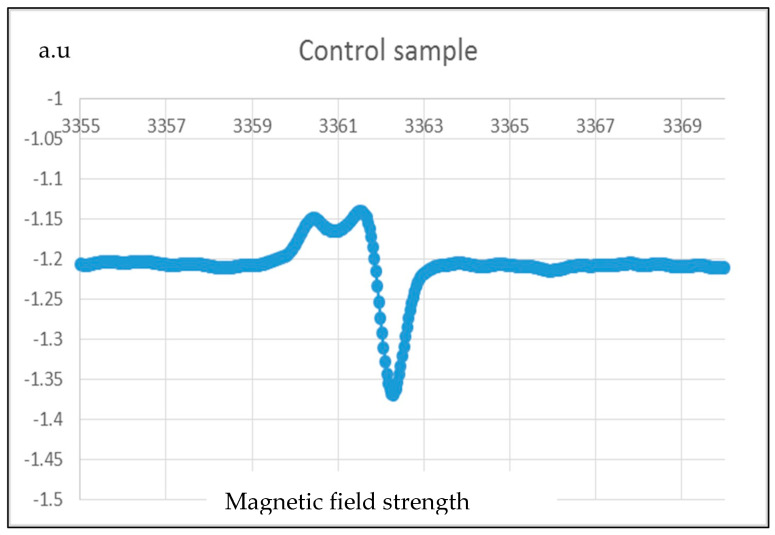
EPR results for control PVDF.

**Figure 18 membranes-13-00231-f018:**
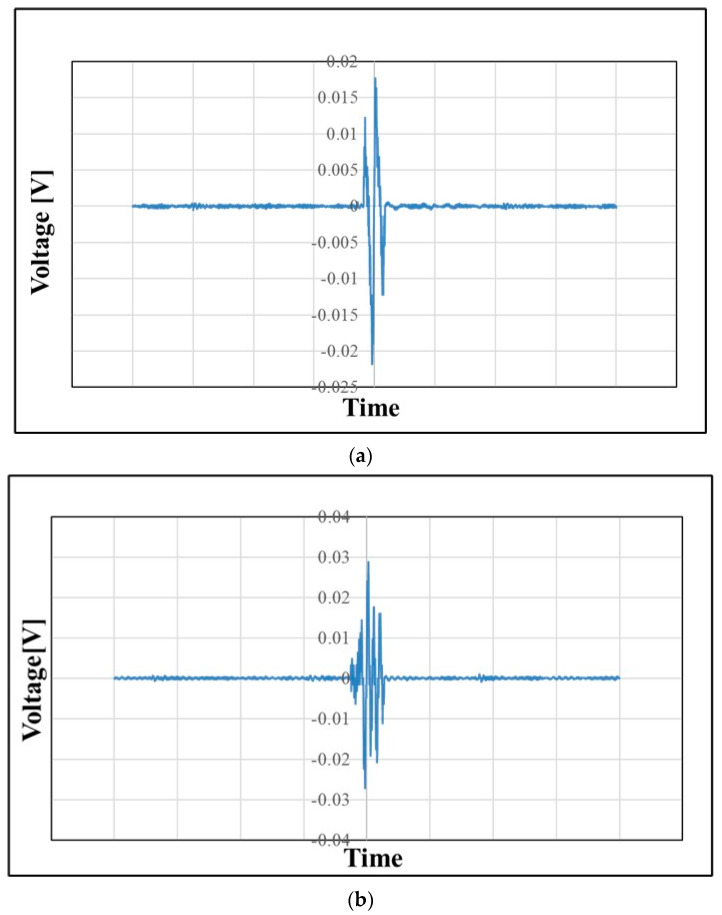
Voltage spike for PAN membrane sensors: (**a**) atmospheric plasma (corona discharge) treated and (**b**) (corona discharge + heat)-treated.

**Figure 19 membranes-13-00231-f019:**
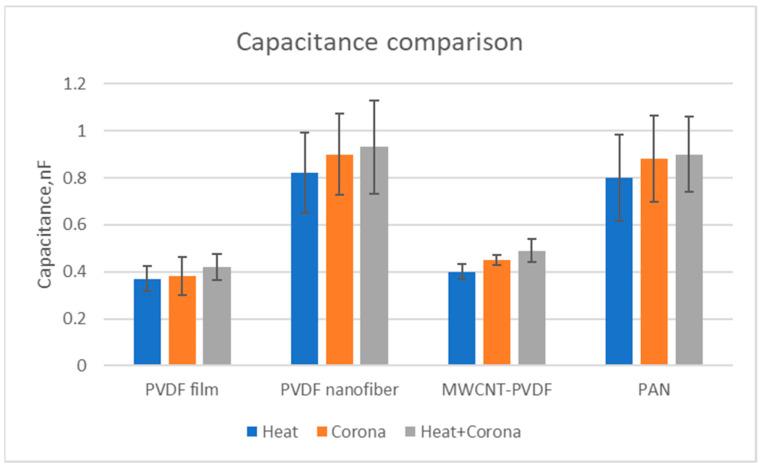
Comparison of capacitance of all nanofiber membrane and films.

**Figure 20 membranes-13-00231-f020:**
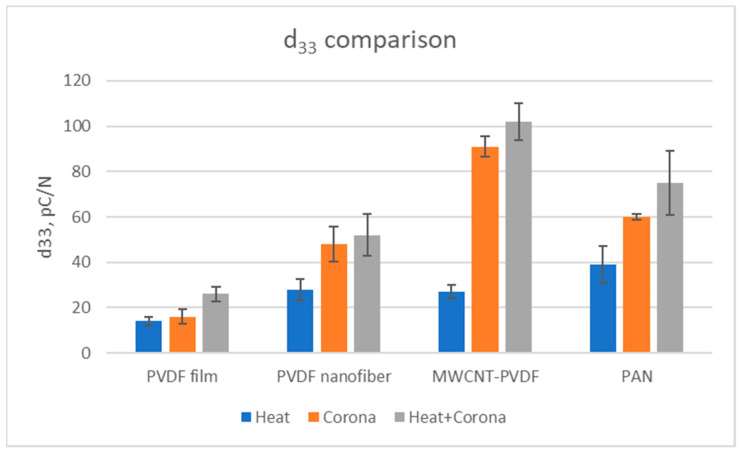
Comparison of d_33_ of all nanofiber membranes and films.

**Table 1 membranes-13-00231-t001:** Sample categories used in the experiments in this work.

Type	Based on Chemical Compositions	Type	Post-Processing Techniques
1.	Film-PVDF	1.	Heat-treated
2.	Nanofiber membranes-PVDF	2.	Corona-treated
3.	Nanofiber membranes-PVDF + 0.2% MWCNTs	3.	Heat + Corona treated
4.	Nanofiber membranes-PAN		

**Table 2 membranes-13-00231-t002:** FTIR results and fraction of β phase formation in PVDF sample.

Control		Corona		Heat		Heat + Corona		Remarks
Wave Number	Absorbance	Wave Number	Absorbance	Wave Number	Absorbance	Wave Number	Absorbance	
840.5	0.01016	840.2	0.02633	840.1	0.02959	840.1	0.02811	**β** phase
762.4	0.0004427	762.3	0.000878	762.5	0.000884	762.4	0.001851	**α** phase
Fraction of β phase content F(β) in PVDF								
0.9479		0.9596		0.9637		0.9233		

**Table 3 membranes-13-00231-t003:** FTIR results and fraction of β phase formation in MWCNT reinforced PVDF sample.

Control		Corona		Heat		Heat + Corona		Remarks
Wave Number	Absorbance	Wave Number	Absorbance	Wave Number	Absorbance	Wave Number	Absorbance	
840.4	0.02867	840.3	0.01184	840.3	0.01639	840.6	0.02349	**β** phase
762.4	0.007529	762.3	0.003329	762.7	0.002555	763.1	0.004784	**α** phase
Fraction of β phase content F(β) in MWCNT-reinforced PVDF								
0.7510		0.7381		0.8356		0.7955		

**Table 4 membranes-13-00231-t004:** Voltage, Capacitance, and piezoelectric coefficient (d_33_) response measurements for Control; Heat treated; Atmospheric plasma (corona discharge) treated and Heat & Corona discharge treated samples.

Sample	PVDF Film			PVDF Nanofiber			MWCNT-Reinforced PVDF			PAN Nanofiber		
	Voltage, mV	Capacitance, nF	d_33_, pC/N	Voltage, mV	Capacitance, nF	d_33_, pC/N	Voltage, mV	Capacitance, nF	d_33_, pC/N	Voltage, mV	Capacitance, nF	d_33_, pC/N
Control	4 ± 0.68	0	0	0	0	0	0	0	0	0	0	0
Heat	12.5 ± 1.52	0.37 ± 0.052	14 ± 1.98	11 ± 2.12	0.82 ± 0.171	28 ± 4.56	22 ± 2.56	0.4 ± 0.032	27 ± 3.1	16 ± 3.25	0.8 ± 0.185	39 ± 8.25
Corona	14 ± 2.65	0.38 ± 0.081	16 ± 3.2	17 ± 2.89	0.9 ± 0.172	48 ± 7.65	65 ± 3.12	0.45 ± 0.02	91 ± 4.56	22 ± 4.36	0.88 ± 0.183	60 ± 11.25
Heat + Corona	20 ± 1.65	0.42 ± 0.054	26 ± 3.25	18 ± 3.62	0.93 ± 0.198	52 ± 9.23	67 ± 5.63	0.49 ± 0.048	102 ± 8.24	27 ± 4.52	0.9 ± 0.162	75 ± 14.1

## Data Availability

Not applicable.

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
