# Peer review of "Investigation of the Effect of Atmospheric Plasma Treatment in Nanofiber and Nanocomposite Membranes for Piezoelectric Applications"

_membranes, 2023, doi:10.3390/membranes13020231_

Round 1
Reviewer 1 Report
The manuscript has an interesting subject and a good amount of experiments are performed. The amount of data presented in the manuscript is sufficient for a research paper however, the presentation style, the design, and the order of results in the manuscript need a major revision. here are some points:
1- The investigation aims 4 different chemical compositions and 3 different treatment techniques, it is recommended to place a table after the materials section, that summarizes the chemical composition of samples.
2- It seems that there is a two level relation between the material properties and piezoelectric properties. First the material composition affects the crystallinity and phases of samples and in the next level these parameters affect the piezoelectric properties . However, in the manuscript first the final result of piezoelectric properties are presented and then the changes in crystallinity and phases are presented in the results of XRD and other experiments. This order is recommended to be changed. The reasoning must be based on the two level analysis. This is the point that needs a major revision. Changing the order of presenting the data and providing a consistent reasoning.
3- The diagrams especially the ones extracted from MATLAB, need to have better and uniform style throughout the manuscript also the parameters and the units of them are missing in some diagrams including figures 11, 12, 13, 14, 15, 16, 19, and 20.
Author Response
Response to Reviewer’s comments
We would like to thank the reviewer for his/her thoughtful review of our manuscript. We have carefully taken the comments into consideration in preparing the revised version of our manuscript. Responses to the reviewer’s comments are reported as follows:
- The investigation aims 4 different chemical compositions and 3 different treatment techniques, it is recommended to place a table after the materials section, that summarizes the chemical composition of samples.
Ans: Thanks to the Reviewer for the suggestion. We have included text and a Table in the materials section accordingly.
2- It seems that there is a two-level relation between the material properties and piezoelectric properties. First the material composition affects the crystallinity and phases of samples and in the next level these parameters affect the piezoelectric properties. However, in the manuscript first the final result of piezoelectric properties are presented and then the changes in crystallinity and phases are presented in the results of XRD and other experiments. This order is recommended to be changed. The reasoning must be based on the two-level analysis. This is the point that needs a major revision. Changing the order of presenting the data and providing a consistent reasoning.
Ans: Thanks to the Reviewer for the suggestion. Following the suggestion, we have changed the order of the discussions in the Result and Discussion section of the revised manuscript.
3- The diagrams especially the ones extracted from MATLAB, need to have better and uniform style throughout the manuscript also the parameters and the units of them are missing in some diagrams including figures 11, 12, 13, 14, 15, 16, 19, and 20.
Ans: Most of these Figures are machine/equipment generated. We have enlarged the figures so that the numbers, units, annotations are visible.
Reviewer 2 Report
The author carefully reviewed the outcome of the results in PVDF nanofiber membranes, MWCNT-reinforced PVDF nanofiber and PAN nanofibers were fabricated by electrospinning process and treated with atmospheric Corona discharge. Further, test results were reported after sensors were fabricated from these nanofiber membranes.
Overall, the manuscript is well presented and there are some nice results. The manuscript is well written overall, but several sentences need to be improved with English corrections. So the manuscript is not recommended for publication. Though the work is innovative and it lacks proper experimental design. The review done about the work is inadequate. It is poorly structured and written and also not following the author guidelines.
The following comments need to be clarified:
1. In the abstract, should include the expansion of PVDF, CNT PVDF and PAN
2. Treatment is performed on nanofiber’s surface for 2 h. Why 2 hours?
3. Page 11, last sentence “It has been reported that the heat treatment is the heat treatment can change the crystal structure from cubic to orthorhombic” should be rewrite.
4. XRD analysis is reported for PVDF only. What about other samples?
5. Ref. 8, Year is missing.
6. Should follow the format of each reference as per journal instructions.
Author Response
Response to Reviewer’s comments
We would like to thank the reviewer for his/her thoughtful review of our manuscript. We have carefully taken the comments into consideration in preparing the revised version of our manuscript. Responses to the reviewer’s comments are reported as follows:
- In the abstract, should include the expansion of PVDF, CNT PVDF and PAN
Ans: Thanks to the Reviewer for the suggestion. We have included the expansion of PVDF, CNT PVDF and PAN in the abstract.
- Treatment is performed on nanofiber’s surface for 2 h. Why 2 hours?
Ans: We have performed the atmospheric plasma (corona) treatment in all categories of samples for total 2 hours. It includes both side of the membrane surface. We have optimized the time of charging based on the surface size and the piezoelectric response. In this study we did not focus any variable timing of charging and its effect. It can be a different future study all together. In this study we have focused on the study of the effect of plasma charging on the nanofiber membranes of different materials systems.
- Page 11, last sentence “It has been reported that the heat treatment is the heat treatment can change the crystal structure from cubic to orthorhombic” should be rewrite.
Ans: Thanks for pointing out. We have revised the section and text has been corrected.
- XRD analysis is reported for PVDF only. What about other samples?
Ans: In this study we have focus on the materials morphology/structure’s property due to the heat and corona plasma treatment. In this case we standardize PVDF, since it shows the maximum piezoelectric effect. We plan to study the morphology and structure property relations for other material system in a future study.
- 8, Year is missing.
Ans: It is actually a chapter of a Book, we have added the DOI: https://doi.org/10.1007/978-3-540-68683-5_6
- Should follow the format of each reference as per journal instructions.
Ans: References have been updated
Reviewer 3 Report
This review relates to the manuscript number "membranes-2068443" submitted to the "Membranes" journal entitled "Investigation of the effect of atmospheric plasma treatment in polymeric nanofiber and nanocomposite membranes for piezoelectricity". The work presents the effect of the heat and plasma treatment on PVDF and CNT-modified PVDF membranes to evaluate their piezoelectric properties. Although the discussion on the characterization results of the composites is quite well, the work finds lacks of real applications like LED lighting using the fabricated device. I feel the manuscript can be accepted after alleviating those issues.
Here are the few issues which should be clarified:
1. The author should provide the novelty of using MWCNT in the PVDF matrix, in spite of the other conducting nanofillers.
2. In Figure 3, the colors of the films are different at different conditions. The author should provide the reason.
3. As capacitance is directly related with dielectric constant which is also related with d33. How the author encounters the enhancement in capacitance with the d33 value and voltage output? (https://www.mdpi.com/2673-706X/2/4/14)
4. The capacitance PVDF film and MWCNT-PVDF are almost same but they produce different voltage output. How?
5. In the reference 20 and 21, there is no report on the cubic to orthorhombic crystal structure transformation.
6. The treatment of heat and corona together decrease the β-phase, but they produce highest voltage. The author should explain this.
7. The conclusion part is less informative. The synergistic effect of heat and corona treatment on the crystal structure of PVDF should be mentioned. The author should provide the responsible phase for the higher d33 and voltage output.
8. There are few typing errors like MWCT. The author should take care of this.
Author Response
Response to Reviewer’s comments
We would like to thank the reviewer for his/her thoughtful review of our manuscript. We have carefully taken the comments into consideration in preparing the revised version of our manuscript. Responses to the reviewer’s comments are reported as follows:
- The author should provide the novelty of using MWCNT in the PVDF matrix, in spite of the other conducting nanofillers.
Ans: Thanks to the reviewer for this suggestion. We have provided a new new text in the introduction section in this regard -
“In comparison with other conducting nanofillers, CNT has higher mechanical strength and elasticity and conductivity. Moreover, it has been reported that a graphene sheet demonstrates surface piezoelectricity and flexoelectricity in certain cases such as: i) when non- centrosymmetric pores are formed in it [1,2,3], ii) if a bending moment and biaxial strain [4,5,6] is applied to it, and through the selective surface adsorption of atoms [2,7,8,9]. Also, it is demonstrated that a significant bending moment is initially formed in carbon nanotubes (CNTs), which can lead to the manifestation of the surface piezoelectricity [3]. For these reasons, in this work we wanted to investigate the piezoelectric properties of MWCNT by fabricating the MWCNT-reinforced PVDF nanofibers.
- In Figure 3, the colors of the films are different at different conditions. The author should provide the reason.
Ans: The brightness setting of the images was different, that’s why they look different, we’ve provided a new picture after fixing the brightness in the updated manuscript.
- As capacitance is directly related with dielectric constant which is also related with d33. How the author encounters the enhancement in capacitance with the d33value and voltage output? (https://www.mdpi.com/2673-706X/2/4/14)
Ans: In our experimental investigation, we have presented the effect of 1) different material system 2) heat and corona plasma treatment on the piezoelectric and capacitance response. The piezo electric effect (Voltage and d33) observed to be higher in MWCT-PVDF membrane samples, whereas capacitance was higher in pristine PVDF samples. However, it also is observed that both voltage and d33 have increased in all material systems (PVDF, MWCNT-PVDF, PAN) when treated with heat and corona discharge (Table-2; Figure 6 &7). Regarding the capacitance, it also shows slight increase due to the heat and corona treatment irrespective of material system. We have observed the complex effect in the β-phase, α-phase, γ-phase and crystallinity due to the material system such as MWCNT reinforcement and the effect of heat and corona discharge treatment. The effect of free radical was also investigated through EPR. We believe, what we observed is a complex combined effect of materials system, changes of its phases, effect of free radicals etc. The reference (https://www.mdpi.com/2673-706X/2/4/14) mentioned by the reviewer is for piezoelectric ceramic which is a completely different material system. In the literature it has been reported that dielectric micro-capacitance strategy has been used to enhance piezoelectricity into a poly(vinylidene fluoride) (PVDF) matrix [1].
1. GuoTian at el “Dielectric micro-capacitance for enhancing piezoelectricity via aligning MXene sheets in composites”, Cell Reports physical science; Volume 3, Issue 4, 20 April 2022, 100814
- The capacitance PVDF film and MWCNT-PVDF are almost same but they produce different voltage output. How?
Ans: Though the capacitance of PVDF (0.93 nF) is higher compared to MWCNT-PVDF (0.49nF) nanofiber membranes as can be seen in Table 2 and Figure 6. However, in all the material system Capacitance slightly increased due to heat and Corona treatment. However, both Voltage and d33 was higher in MWCNT-PVDF compared to pristine PVDF nanofiber membranes. In this case, volage response and d33 increased when treated with heat and corona. The trend is observed to be same for all material system when treated with heat and corona. We have investigated different materials system as well as the effect of heat and corona discharge. In Raman spectroscopy results, we’ve seen presence of higher β-phase in MWCNT-reinforced nanofibers. Also, from the EPR, we compared control sample and corona-treated sample and observed increase of free radicals in the corona-treated samples.
- In the reference 20 and 21, there is no report on the cubic to orthorhombic crystal structure transformation.
Ans: Thank you for pointing that out, this portion of the manuscript has been corrected.
- The treatment of heat and corona together decrease the β-phase, but they produce highest voltage. The author should explain this.
Ans: There were two-fold effect in the piezoelectric response. 1) the materials system, 2) effect of heat and corona discharge. In the FT-IR results, we’ve seen a slight decrease in β-phase fraction in Heat+Corona treated sample; however, β-phase has increased in the MWCNT-PVDF material system compared to pristine PVDF samples. Also, increased γ-phase is observed due to the heat and corona treatment. Moreover, EPR tests showed increased free radical due to the corona discharge. We believe it is an complex combined effect of multiple phenomena.
- 7. The conclusion part is less informative. The synergistic effect of heat and corona treatment on the crystal structure of PVDF should be mentioned. The author should provide the responsible phase for the higher d33and voltage output.
Ans: We’ve rewritten the conclusion part in the updated manuscript.
In this work, PVDF nanofiber membranes, MWCNT-reinforced PVDF nanofibers and PAN nanofibers were fabricated by electrospinning process and treated with atmospheric steady-state plasma (Corona discharge). Sensors were fabricated from these nanofiber membranes and tested. In all categories of samples, Piezoelectric coefficient (d33) showed an increasing trend in heat and corona-treated samples. The d33 was almost two times greater in corona and heat-treated MWCNT-reinforced PVDF samples than the treated PVDF samples. Treated PVDF film samples exhibited 20mV output voltage, while the treated PVDF nanofibers showed 65mV output voltage. The d33 was observed to be higher in nanofiber membranes when compared with the PVDF film. XRD results showed that the α-phase found in Control sample, shifted slightly in all the treated samples towards a greater angle from 16.3⁰ to 17⁰. In the FT-IR characterization of PVDF samples, we found a peak of α-phase in 796.0cm-1 in the control sample, which was not prominent in the treated samples. A peak in 1233cm-1 corresponds to the -phase was found in all the samples. We also found peaks at 763cm-1, 975cm-1 corresponding to α-phase and peaks at 840cm-1, 1275cm-1 corresponding to β-phase in all the samples. There was an increase of α-phase peaks in MWCNT-reinforced PVDF samples compare to the only PVDF samples. In the MWCNT-reinforced PVDF samples, we found peaks at 763cm-1, 797cm-1, 975cm-1 corresponding to α-phase and 840cm-1, 1275cm-1 corresponding to β-phase. From Raman spectroscopy, it was observed in the PVDF samples, that characteristics α-phase at 788cm-1 existed, and there was a right-shift of this peak from control to the treated sample. The characteristic peaks of CNT (1351cm-1,1597cm-1) was observed in MWCNT-reinforced PVDF samples. We also found peaks at 1598 cm-1, 1350 cm-1, 2688 cm-1, 2979 cm-1 corresponding to G-band, peak at 1435 cm-1 corresponding to D-band and peaks at 614cm-1, 842cm-1 corresponding to β-phase. All the shift observed in MWCNT-reinforced PVDF samples were right-shift. In the EPR characterization, we compared control sample and corona-treated sample and observed that 8% increase of free radicals in the corona-treated sample. Thus, it can be concluded, that corona discharge treatment increases the quantity of free radicals which eventually was evident in the piezoelectric properties of the samples. Hence, all the characterization techniques demonstrated the effect of atmospheric plasma treatment in the enhancement of piezoelectric properties of nanofibers and nanocomposites. Utilizing the piezoelectric properties of PVDF nanocomposites as sensors, a wide range of applications in medical diagnostics, wearable systems, structural health monitoring system, electromechanical equipment etc. can be achieved.
- There are few typing errors like MWCT. The author should take care of this.
Ans: Thank you, the manuscript has been corrected.
References:
- Il’ina MV, Il’in OI, Rudyk NN, Osotova OI, Fedotov AA, Ageev OA. Analysis of the piezoelectric properties of aligned multi-walled carbon nanotubes. 2021,11(11), 2912.
- Javvaji, B.; He, B.; Zhuang, X. The generation of piezoelectricity and flexoelectricity in graphene by breaking the materials symmetries. Nanotechnology 2018, 29, 225702.
- Chandratre, S.; Sharma, P. Coaxing graphene to be piezoelectric. Phys. Lett. 2012, 100, 023114.
- Wang, X.; Tian, H.; Xie, W.; Shu, Y.; Mi, W.-T.; Mohammad, M.A.; Xie, Q.-Y.; Yang, Y.; Xu, J.-B.; Ren, T.-L. Observation of a giant two-dimensional band-piezoelectric effect on biaxial-strained graphene. NPG Asia Mater. 2015, 7, e154.
- El-Kelany, K.E.; Carbonniere, P.; Erba, A.; Sotiropoulos, J.-M.; Rérat, M. Piezoelectricity of Functionalized Graphene: A QuantumMechanical Rationalization. Phys. Chem. C 2016, 120, 7795–7803.
- Duggen, L.; Willatzen, M.; Wang, Z.L. Mechanically Bent Graphene as an Effective Piezoelectric Nanogenerator. Phys. Chem. C 2018, 122, 20581–20588.
- Ong, M.T.; Reed, E.J. Engineered Piezoelectricity in Graphene. ACS Nano 2012, 6, 1387–1394.
- Bistoni, O.; Barone, P.; Cappelluti, E.; Benfatto, L.; Mauri, F. Giant effective charges and piezoelectricity in gapped graphene. 2D Mater. 2019, 6, 045015.
- Ong, M.T.; Duerloo, K.-A.N.; Reed, E.J. The Effect of Hydrogen and Fluorine Coadsorption on the Piezoelectric Properties of Graphene. Phys. Chem. C 2013, 117, 3615–3620.
Reviewer 4 Report
This work has reported the effect of steady-state atmospheric plasma in electrospinning membranes for piezoelectric application. In general, I think this work is less innovative, and some issues must be addressed before resubmit.
1、 Manuscript should review and complement recent works in the same research fields.
2、 What is the innovation of this work? What are the differences from other research works in the same research fields?
3、 There are only results of the experiments, but no discussions. e.g., MWNT reinforced PVDF nanofiber membranes show less capacitance when compared with PVDF nanofiber membrane. What is the reason or mechanism for this result.
4、 There should be more characterization and analysis of piezoelectric properties in manuscript.
Author Response
Response to Reviewer’s comments
We would like to thank the reviewer for his/her thoughtful review of our manuscript. We have carefully taken the comments into consideration in preparing the revised version of our manuscript. Responses to the reviewer’s comments are reported as follows:
- Manuscript should review and complement recent works in the same research fields.
Ans: We would like to thank the reviewer for the suggestion. We have added text/paragraph in regard to the recent related work/literature in the introduction section.
In comparison with other conducting nanofillers, CNT has higher mechanical strength and elasticity. Moreover, it has been reported that a graphene sheet demonstrates surface piezoelectricity and flexoelectricity in certain cases such as: i) when non- centrosymmetric pores are formed in it [12-14], ii) if a bending moment and biaxial strain [15-17] is applied to it, and through the selective surface adsorption of atoms [13,18-20]. Also, it is demonstrated that a significant bending moment is initially formed in carbon nanotubes (CNTs), which can lead to the manifestation of the surface piezoelectricity [14]. For these reasons, in this work we wanted to investigate the piezoelectric properties of MWCNT by fabricating the MWCNT-reinforced PVDF nanofibers.
2、 What is the innovation of this work? What are the difference from other research works in the same research fields?
Ans: The innovation of this work is that in specific scale and materials system atmospheric plasma treatment can induce piezoelectric effect as well as have effect on materials capacitance. Such work has engineering and applied importance on flexible sensor materials and multifunctional fibers and membranes fabrications. In our experimental investigation, we have presented the effect of 1) different material system 2) heat and corona plasma treatment on the piezoelectric and capacitance response. The piezo electric effect (Voltage and d33) observed to be higher in MWCT-PVDF membrane samples, whereas capacitance was higher in pristine PVDF samples. However, it also is observed that both voltage and d33 have increased in all material systems (PVDF, MWCNT-PVDF, PAN) when treated with heat and corona discharge. We have also elaborately discussed the structure property relations/changes using XRD, Raman Spectroscopy, FTIR, EPR analysis caused by the effect of the heat and atmospheric plasma treatment on pristine and nanocomposite structures.
To our knowledge PVDF has been studied for piezoelectricity but not the enhancement of its Piezoelectricity due to plasma interaction. Also we report the effect of material system specifically on nanocomposite and charging with plasma can have profound impact on piezoelectricity and capacitance on flexible polymeric materials at nanoscale.
3、 There are only results of the experiments, but no discussions. e.g., MWNT reinforced PVDF nanofiber membranes show less capacitance when compared with PVDF nanofiber membrane. What is the reason or mechanism for this result.
Ans: Regarding the capacitance, shows slight increase due to the heat and corona treatment irrespective of material system. We have observed the complex effect in the β-phase, α-phase, γ-phase and crystallinity due to the material system such as MWCNT reinforcement and the effect of heat and corona discharge treatment. The effect of free radical was also investigated through EPR. We believe, what we observed is a complex combined effect of materials system, changes of its phases, effect of free radicals etc. Details observations are mentioned in the Raman, FTIR, EPR and XRD sections.
4、 There should be more characterization and analysis of piezoelectric properties in manuscript.
Ans: We would continue to work more on characterizations for future study and publications.
Round 2
Reviewer 1 Report
Due to the amendments and the responses, the manuscript is suitable for being published.
Reviewer 2 Report
After careful review of this article, it can accept for publication.
Reviewer 3 Report
Tha authors have addressed all the queries successfully. The manuscript should be accepted in its current form.
Reviewer 4 Report
The authors have adressed all my questions for this manuscript, and this manuscript has been significantly improved after revising.